# Non-Antimicrobial Adjuvant Strategies to Tackle Biofilm-Related *Staphylococcus aureus* Prosthetic Joint Infections

**DOI:** 10.3390/antibiotics10091060

**Published:** 2021-09-01

**Authors:** Narayan Pant, Damon P. Eisen

**Affiliations:** College of Medicine and Dentistry, James Cook University, Townsville, QLD 4811, Australia; damon.eisen@jcu.edu.au

**Keywords:** non-antimicrobial adjuvant, biofilm, *Staphylococcus aureus*, prosthetic joint infections

## Abstract

*Staphylococcus aureus* frequently causes community- and hospital-acquired infections. *S. aureus* attachment followed by biofilm formation on tissues and medical devices plays a significant role in the establishment of chronic infections. Staphylococcal biofilms encase bacteria in a matrix and protect the cells from antimicrobials and the immune system, resulting in infections that are highly resistant to treatment. The biology of biofilms is complex and varies between organisms. In this review, we focus our discussion on *S. aureus* biofilms and describe the stages of their formation. We particularly emphasize genetic and biochemical processes that may be vulnerable to novel treatment approaches. Against this background, we discuss treatment strategies that have been successful in animal models of *S. aureus* biofilm-related infection and consider their possible use for the prevention and eradication of biofilm-related *S. aureus* prosthetic joint infection.

## 1. *Staphylococcus aureus* Biofilms: Slow Growing Organisms Highly Resistant to Drugs

Approximately 20% and 60% of healthy adults are persistent and intermittent *S. aureus* nasal carriers, respectively [1]. *S. aureus* carriers are at higher risk of endogenous infection. It has been shown that 80% of cases of severe invasive infections in *S. aureus* carriers are caused by strains colonizing their anterior nares [2]. Acute infections are caused by planktonic bacterial forms characterized by free floatation, while sessile forms mainly cause chronic infections [3]. Sessile forms of bacteria are metabolically less active than planktonic forms and are protected by a filmy layer of “slime” referred to as the extra-polymeric substance (EPS) [4]. These properties of sessile forms of bacteria, particularly when they form biofilms, make them recalcitrant to antibiotic treatment [4,5]. Biofilms represent a mode of bacterial growth that acts as a multi-cellular structure, where each bacterial cell works in coordination to keep the structure alive and safe from adverse conditions [6].

## 2. Stages of Biofilm Formation in *S. aureus*

Bacterial biofilm formation occurs in three sequential stages: 1. attachment, 2. maturation, and 3. dispersal. Free-floating planktonic cells attach to surfaces and multiply to form micro-colonies. During maturation, these micro-colonies produce an extracellular matrix and form solid three-dimensional biofilm structures. After full maturation of the biofilm, the extracellular matrix degrades and releases part of the bacteria to establish a new biofilm at another location (Figure 1).

### 2.1. The First Step: Attachment of S. aureus to Surfaces

Planktonic cells come into contact with surfaces with the help of gravitational forces and Brownian movement [7]. Attracting and repelling forces arising from physicochemical and electrostatic interactions between bacterial cells and inanimate surfaces cause the initial and reversible bacterial attachment [8]. Negatively charged extracellular DNA (eDNA) helps to develop an electrostatic interaction [1]. In *S. aureus*, microbial surface components, such as fibronectin-binding proteins (FnBPA and FnBPB), clumping factors, and Protein A, referred to collectively as microbial surface components recognizing adhesive matrix molecules (MSCRAMMs), play an important role in the initial attachment to surfaces [9,10,11,12]. *S. aureus* expresses up to 24 different cell-wall-anchored proteins, including MSCRAMMs (FnBPs, ClfB, and SdrC proteins) and other proteins, such as SasG, Bap, and SasC (Table 1) [13]. These intrinsic matrix molecules attach to the *S. aureus* cell wall after being cleaved by the membrane-associated protein Sortase A [14], and they interact with host matrix components, such as cytokeratin, fibronectin, collagen, and fibrinogen [15]. Loss of the Sortase and mutations in the *fnbA* and *fnbB* genes encoding FnBPA and FnBPB, respectively, reduces biofilm formation in methicillin-resistant *S. aureus* (MRSA) [10]. Similarly, mutants of the *S. aureus* Newman strain defective in clumping factor A adhere poorly to fibrinogen-coated polymethylmethacrylate (PMMA) coverslips and do not form clumps in soluble fibrinogen [11]. Additionally, novel staphylococcal Protein A receptor, C1qR, has been identified in wounds [12].

MSCRAMMs play a lesser role in attachment to abiotic surfaces, where electrostatic and hydrophobic interactions predominate in the initial attachment [16]. Apart from MSCRAMMs, teichoic acid, a negatively charged component of the *S. aureus* cell wall, is also responsible for the initial, relatively loose attachment of planktonic cells [17]. Therefore, while there are multiple mechanisms contributing to initial attachment, the process is a dynamic one and bacteria may detach in response to repulsive forces and limited nutrient availability in biofilms including iron [8,18].

### 2.2. Maturation of the S. aureus Biofilm

After initial attachment, and in the presence of sufficient nutrients, bacteria begin to form micro-colonies [19]. Concurrently, changes in gene expression are triggered in response to surface contact, leading to the upregulation of factors favoring transformation into sessile forms [20]. As these micro-colonies grow, they produce EPS to form a mature biofilm [21]. Biofilm maturation is characterized by intercellular aggregation and three-dimensional structure formation [22]. In *S. aureus*, polysaccharide intercellular adhesin/poly-β(1–6)-N-acetylglucosamine (PIA/PNAG) is responsible for intercellular aggregation [23]. Intercellular adhesion (*ica*) locus mutant does not produce PIA/PNAG and hence biofilm [23]. PIA/PNAG in combination with teichoic acid and proteins contributes to the extracellular matrix of staphylococcal biofilm [22].

Expression of cell-wall-anchored proteins varies among strains and growth conditions as some proteins are expressed only in iron-deficient conditions, while others are preferentially expressed at exponential or stationary growth phases [13]. These proteins facilitate intercellular binding and therefore the accumulation of bacterial cells [13]. For example, *S. aureus* strains expressing the biofilm-associated protein (Bap) showed strong intercellular and surface adherence resulting in luxuriant in vitro biofilm formation and persistent infection in a mouse infection model, in contrast to *bap* mutants, which were less adherent and weak biofilm producers [24]. Similarly, the addition of Protein A to growth media induced biofilm formation, which was completely inhibited after the addition of antibodies against Protein A [25]. In a murine model of subcutaneous catheter infection, the number of wild-type bacteria recovered was significantly higher than the Protein A-deficient bacteria, when the medical implant was co-infected with both the strains [25].

*S. aureus* also uses cytoplasmic proteins, such as enolase and GAPDH, as matrix components [26]. These cytoplasmic proteins, probably released through autolysis, attach to cell surfaces and eDNA at low pH and help in the formation of a stable three-dimensional biofilm structure [26,27,28]. However, an *S. aureus* biofilm formation model in which the bacteria do not use dedicated biofilm matrix proteins but recycle cytoplasmic proteins released at the stationary phase has been proposed [26]. Other mechanisms for cytoplasmic protein release may be secretion, vesicle formation, and bacteriophage-related cell lysis [29]. Extracellular proteins such as phenol-soluble modulins (PSMs), [30] and nucleoid-associated proteins also help in biofilm stabilization by binding with eDNA [31], an important structural component of the mature staphylococcal biofilm [32].

### 2.3. Triggering of the Biofilm Dispersal Response

Following biofilm maturation, bacterial cells disperse to start a new cycle of biofilm formation at distant sites [33]. In *S. aureus* biofilms, early dispersal may begin after six hours through the nuclease-dependent degradation of eDNA [34]. This early dispersal is known as ‘exodus’ and helps in biofilm reorganization [34]. Exodus involves a subpopulation of biofilm cells that secret nuclease [34]. 

Later stages of *S. aureus* biofilm dispersal are orchestrated by the *agr* quorum sensing (QS) system (Figure 2) [35]. Quorum sensing (QS) is a coordinated cell to cell communication induced by chemical signals [36]. In *S. aureus*, these signals are short cyclical peptides known as auto-inducing peptides (AIPs) [37]. The *S. aureus agr* system consists of four genes (*agrA*, *agrB*, *agrC*, *agrD*), among which *agrD* and *agrB* synthesize and export AIPs, while *agrC* and *agrA* form a signal transduction system [37,38]. On the accumulation of extracellular AIPs to threshold level, they bind to and activate histidine kinase, AgrC, which then phosphorylates AgrA, and, in turn, AgrA binds to promoters P2 and P3, and finally, regulatory molecules, RNA II and RNA III, respectively, are expressed [37,39]. RNA II encodes components of the *agr* system, i.e., AgrB, AgrD, AgrC, and AgrA [37], while RNA III encodes several other *S. aureus* virulence factors [40,41]. In addition, P3 activation increases protease activity through extracellular protease production, which contributes to the degradation of the protein-based biofilm matrix [42].

Alternatively, *agr*-dependent dispersal may also occur through the production of PSMs, which have surfactant properties and cause biofilm dispersal by interacting with the biofilm matrix [43,44]. These modulins are produced when phosphorylated AgrA binds to the *psm* operon promoter region [44]. However, PSM aggregates can also stabilize biofilm structures through insoluble amyloid fiber production [45]. Formation of these amyloid fibers is promoted by the presence of eDNA [30]. Hence, the role of PSM in biofilms depends upon the state in which it is produced.

**Table 1 antibiotics-10-01060-t001:** *S. aureus* biofilm components and their functions.

Biofilm/Cell Components	Biofilm Stages	Functions	References
eDNA	Attachment	Development of electrostatic interaction for initial attachment	[1]
	Maturation	Biofilm matrix formation and biofilm stabilization	[28]
Cell-wall-anchored proteins	Attachment	Initial attachment	[9,10,11,12,13]
	Maturation	Intercellular binding and bacterial cell accumulation	[13]
Sortase A	Attachment	Cleavage of cell-wall-anchored proteins to catalyze initial attachment	[14]
Teichoic acid	Attachment	Initial attachment	[17]
	Maturation	Biofilm matrix formation	[22]
Cytoplasmic proteins	Maturation	Biofilm matrix formation and biofilm stabilization by binding with eDNA	[26]
PSMs	Maturation	Biofilm stabilization by forming insoluble amyloid fibers and binding with eDNA	[30,45]
	Dispersal	Biofilm dispersal by interacting with biofilm matrix	[43,44]
Nucleoid-associated proteins	Maturation	Biofilm stabilization by binding with eDNA	[31]
Nucleases	Dispersal	Biofilm dispersal through degradation of eDNA	[34]
Proteases	Dispersal	Biofilm dispersal through degradation of protein component of biofilm	[42]
AIPs	Dispersal	Biofilm dispersal through activation of *agr* quorum sensing system	[37,39]

**Figure 2 antibiotics-10-01060-f002:**
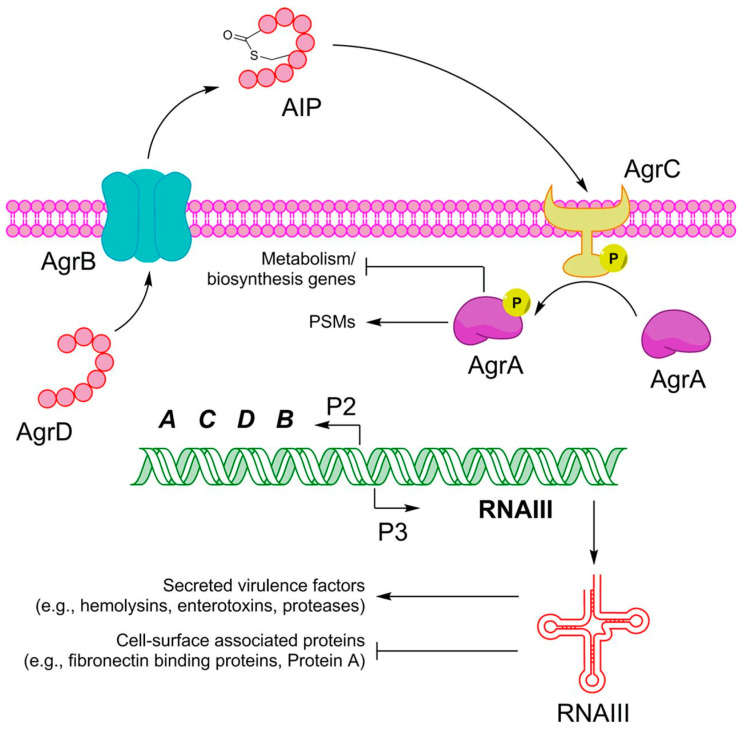
*S. aureus* accessory gene regulatory (*agr*) system. Briefly, AgrD and AgrB synthesize and export AIPs to external environment. On accumulation of extracellular AIPs to threshold levels, they bind to and activate histidine kinase, AgrC, which then phosphorylates AgrA. AgrA then binds to different promoter regions, driving expression of components of *agr* system, *S. aureus* virulence factors, and PSMs. (Reproduced from https://msphere.asm.org/content/msph/3/1/e00500-17/F1.large.jpg (accessed on 5 April 2021) by Salam A.M., Quave C.L., Targeting virulence in *Staphylococcus aureus* by chemical inhibition of the accessory gene regulator system in-vivo, mSphere, 2018, by permission from copyright holder Salam and Quave, 2018, under creative commons license https://creativecommons.org/licenses/by/4.0/ (accessed on 5 April 2021) [46]).

## 3. Biofilm Formation through PIA/PNAG-Dependent Mechanism

Production of PIA/PNAG is controlled by the *ica* operon, which is upregulated in anaerobic conditions, such as inside biofilms (Figure 3) [23,47]. Under anaerobic conditions, SrrAB, a staphylococcal respiratory response regulator, induces PIA/PNAG production by the binding of phosphorylated SrrA to the *ica* operon promoter region [48]. However, the production of PIA/PNAG may also be induced by other adverse environmental conditions, such as excess glucose, sub-inhibitory antibiotic concentrations, high osmolarity, and temperature [49]. The stress response in *S. aureus* is regulated by Spx, which downregulates biofilm formation by modulating IcaR, a negative regulator of *icaADBC* [50]. However, Rbf, a regulator of biofilm formation, represses *icaR* and enhances biofilm formation by increasing *ica* expression and PIA production [51]. TcaR, a teicoplanin-associated locus transcriptional regulator, can also repress PIA synthesis [52]. However, TcaR is a weaker negative regulator than IcaR, because *icaR* expression can mask the phenotypic effect of *tcaR* deletion [52]. Therefore, it can be concluded that most of the *ica* regulators rely on IcaR to regulate PIA/PNAG-dependent biofilm formation.

Insertion sequence (*IS256*) [53] and a two-component *ica* repressor system, *arLRS*, are other *ica* operon regulators [54]. Insertion of *IS256* inactivated the *icaC* gene and converted a biofilm-positive *S. aureus* strain into a biofilm-negative phase variant, reducing bacterial adherence to surfaces, the preliminary step for biofilm formation [53]. In contrast, initial attachment and PIA/PNAG accumulation are enhanced in the *arlRS* mutant, and biofilm formation is not affected by deletion of the *icaADBC* operon [54]. This indicates the presence of alternative mechanism of biofilm formation.

## 4. Biofilm Formation through PIA-Independent Mechanisms

*ica* locus deletion had no effect on biofilm formation by MRSA strain, BH1CC, while other mutant strains lost their biofilm-producing ability [55]. PIA-independent biofilms consist of eDNA and a long list of proteins that include surface adhesins, secreted proteins, and intracellular proteins released during cell lysis (Figure 3) [1,56]. In the absence of PIA, protein A (SpA) is an important component of *S. aureus* biofilm [25]. Surface adhesin FnBp also contributes to biofilm formation with the help of major autolysin (Atl) and *sigB* regulation [57], while secreted proteins, Eap and beta toxin (Hlb), help in mature biofilm establishment [58,59]. Eap is a predominant protein present in the biofilm matrix of *S. aureus* MR23 [59].

The Hlb and DNABII family of proteins, after binding with eDNA, form insoluble components that help to give a three-dimensional structure to the biofilm [31,58]. Extracellular matrix binding protein (Emp) and Eap contribute significantly to *S. aureus* biofilm formation in iron-restricted environment, representative of the in vivo infection [60]. Under iron-deficient conditions, these proteins are regulated by the iron regulator, Fur (ferric uptake regulator) [60]. In addition, *sae*, *agr*, and *ica* are essential for the expression of Eap and Emp, while *sarA* has a less significant role [60]. However, iron regulation of these secreted proteins is Fur-independent [60].

Extracellular DNA helps in the maturation of biofilms and the initial establishment of a *atl*/*fnbp*-dependent biofilm [61]. Earlier, eDNA in biofilms was thought to be excreted through membrane vesicles rather than cell death [62]. However, later genomic DNA release was demonstrated to occur via *cidA*-controlled cell lysis [32]. A *cidA* mutant exhibited a less adherent and moderately DNase I-sensitive biofilm, with more dead cells accumulated—indicative of reduced cell lysis, and five-fold less genomic DNA in comparison to the parental strain’s (UAMS-1) highly DNase I-sensitive biofilm [32]. The *cid* operon upregulates *atl* and *lytM*, leading to the production of murein hydrolases, which are responsible for bacterial autolysis [63]. This autolysis is induced in certain biofilm micro-environments, such as hypoxic conditions [64]. *cidA*-controlled cell lysis is downregulated through activation of the *lrgAB* operon by LytSR, a two-component regulatory system [61,65]. A *lytS* mutant produced luxuriant biofilm containing a higher amount of matrix-associated eDNA relative to the parent strain [65]. *cid* and *irg* operons work in a way close to bacteriophage holins and anti-holins [63]. Holins and anti-holins are membrane proteins that regulate bacteriophage-induced bacterial death and lysis [66]. Phages have been detected in biofilm culture with the help of an electron microscope [67]. Activation of phage genes may also help in the release of *S. aureus* DNA through cell lysis, leading to phage release [67]. However, these studies suggest that the mechanism of eDNA release is strain-specific as different modes of eDNA release were found in different *S. aureus* strains.

Amyloid and fibrin are other types of PIA-independent *S. aureus* biofilms [45,68]. Amyloid biofilms consist of amyloid fibers formed from PSM aggregates [45] and the formation of these fibers is promoted by the presence of eDNA [30]. A fibrin biofilm is formed on plasma-coated surfaces, where coagulase (Coa) from *S. aureus* converts fibrinogen to fibrin [68]. Fibrin thus formed makes a scaffold for an *S. aureus* biofilm [69]. The *saeRS* system regulates *coa* expression, thus taking part in the formation of a fibrin-mediated biofilm [70]. 

**Figure 3 antibiotics-10-01060-f003:**
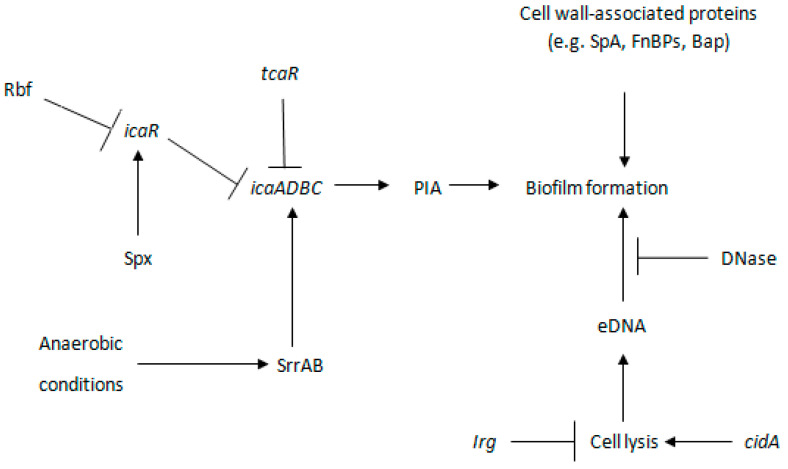
PIA-dependent and -independent *S. aureus* biofilm formation. Briefly, Rbf downregulates *icaR*, while Spx upregulates it. *icaR* and *tcaR* downregulate *ica*. Under anaerobic conditions, regulator SrrAB upregulates *ica*, leading to PIA production and PIA-dependent biofilm formation. Cell-wall-associated proteins and eDNA released through *cidA*- and *irg*-regulated cell lysis form PIA-independent biofilms. *irg* downregulates cell lysis while *cidA* upregulates it. eDNA biofilm formation is prevented by DNase. (Adapted from https://www.ncbi.nlm.nih.gov/pmc/articles/PMC3322633/figure/F5/?report=objectonly (accessed on 5 April 2021) by Archer N.K., Mazaitis M.J., Costerton J.W., et al., *Staphylococcus aureus* biofilms: properties, regulation, and roles in human disease, Virulence, 2011, by permission from copyright holders Taylor & Francis, 2011, under creative commons license https://creativecommons.org/licenses/by-nc/3.0/ (accessed on 5 April 2021) [71]).

## 5. Regulation of *S. aureus* Biofilm Formation: The Master Controllers and Their Targets

Biofilm formation in *S. aureus* is controlled by *sarA*, *agr*, *sigB*, and *sae* regulons [42,72,73,74] (Figure 4). *sarA* and *agr* regulate a two-component virulence regulator system, *arlS-arlR* [75]. This system downregulates the production of virulence factors, such as beta-hemolysin, alpha-toxin (Hla), serine protease (Ssp), lipase, coagulase, and Spa [75]. Mutations in either component of *arlS-arlR* enhance the secretion of these proteins [75].

### 5.1. sarA

*sarA* upregulates *ica*, leading to PIA/PNAG production and consequently increased biofilm formation [76]. Additionally, *sarA* downregulates the expression of a protein that either degrades PIA/PNAG or represses the production; *sigB* upregulates the protein synthesis [76]. Transcriptional profiling suggested that the expression of *fnbA* and *fnbB* is *sarA*-dependent [77]. *sarA* mutants showed reduced biofilm formation in six out of eight *S. aureus* strains tested [78], which can be recovered by *nuc* gene deletion or/and protease inhibition [79]. In *sarA* mutants, there is increased production of extracellular nucleases and proteases that degrade biofilm components [79].

### 5.2. agr

Presence of the *sarA* gene is required for optimal *agr* expression [80]. In *sarA* mutants, the level of *agr* regulatory molecule, RNAIII, was found to be significantly reduced or absent, but it was partially restored when the intact *sarA* gene was reinserted [80]. However, involvement of the *sarA* in biofilm formation is *agr*-independent and *sarA* mutants show reduced biofilm formation despite the functional status of the *agr* [81]. An inactive *agr* quorum sensing system is required for protein-based *S. aureus* biofilm formation, and activation of this system by the addition of AIP or glucose depletion in mature biofilm leads to dispersal [42]. Depending upon strains and growth conditions, the role of the *agr* quorum sensing system may vary, as disruption of the *agr* inhibits, enhances, or has no effect on biofilm formation [82,83]. Another *S. aureus* quorum sensing system, *luxS*, reduces biofilm formation by disrupting intercellular aggregation through downregulation of exopolysaccharide production [84].

### 5.3. sigB

*sigB*, another regulator of *S. aureus* biofilm formation, positively affects the expression of two microbial surface proteins, FnbA and ClfA, which are responsible for the initiation of biofilm formation [85]. *sigB* mutant strain, BB1591, had two-fold lower capacity to be internalized by osteoblasts in comparison to its parent strain, LS-1 [85]. In addition, based on the levels of *sigB* expressed, individual strains of *S. aureus* had differing capacities to be internalized [85]. These authors suggested that *sigB* may increase the expression of MSCRAMMs, such as FnBPs, which play an important role during the internalization of *S. aureus* by osteoblasts [85]. Additionally, *sigB* suppresses *agr* and inhibits biofilm dispersal [86]. In the *sigB* mutant strain USA300 LAC, *agr* RNAIII levels were elevated, which is responsible for biofilm dispersal through the elevation of extracellular protease levels [86]. Similarly, in the COL strain, thermo-nuclease, an enzyme that promotes biofilm dispersal through degradation of eDNA, production was enhanced in a *sigB* mutant compared with the parent strain [61,87]. This higher production of thermo-nuclease suggests an inhibitory role of *sigB* on either the production or excretion of the protein, thus favoring biofilm development [87]. However, the role of *sigB* on thermo-nuclease production is strain-specific, as there was no difference in thermo-nuclease production between *sigB* mutants and the wild type in strains Newman and 8325 [87].

There was no effect of *sigB* deletion in PIA/PNAG-dependent biofilm formation [76], suggesting that *sigB* is directly involved in the regulation of protein biofilms but not in PIA/PNAG-dependent biofilms. However, in the *sarA-sigB* double mutant, *ica* expression decreased but PIA/PNAG production and biofilm formation increased in comparison with the *sarA* single mutant [76]. This means that some indirect role of *sigB* in PIA/PNAG-dependent biofilm formation may exist. Some researchers have reported a loss of PIA/PNAG production and biofilm formation in *sigB* mutants under osmotic stress, suggesting a role of *sigB* in *S. aureus* biofilm regulation under adverse conditions such as heat shock, alkaline shock, high salt, and stationary phase growth in complex media such as LB [88,89].

In the stationary phase of strains LAC, Newman, and 8325, *sigB* mutants showed increased lipase production in comparison with the wild type, suggesting an inhibitory role of the *sigB* on lipase, effecting the biofilm formation negatively [87]. Lipase-coding gene mutants produce weak biofilms in comparison to wild-type strains [90], and biofilm formation can be inhibited by the addition of anti-lipase serum [91]. In addition, intra-peritoneally injected lipase mutant produced a defective peritoneal abscess in mice with a lower concentration of bacteria in different organs in comparison with the wild type [90]. Immunization of mice with recombinant lipase saved mice from lethal *S. aureus* infection [90]. *sigB* is regulated by positive regulator RsbU, negative regulator RsbW, and anti-RsbW regulator RsbV [89].

### 5.4. saeRS

*saeRS*, a two-component *S. aureus* regulatory system, inhibits biofilm formation by producing a heat-stable inhibitory protein that affects the attachment step [74]. In a *sae* constitutively expressed *S. aureus* Newman strain, a weak biofilm producer, deletion of *saeRS* resulted in the production of a robust biofilm [74]. *saeRS* consists of the SaeS protein, a histidine kinase, which phosphorylates SaeR, a response regulator [74]. *sae* upregulates *atlR* and bacteriophage genes; *atlR* encodes a repressor of *atlA,* thus reducing autolysis and DNA release [57,74].

## 6. Clinical Context of Biofilm-Related *S. aureus* Prosthetic Joint Infections: Failure of Life-Enhancing Prosthetic Joints

Medical devices, such as prosthetic joints, provide a surface for bacterial proliferation and biofilm formation [92]. Every year, around two million total knee and hip arthroplasties are carried out worldwide [93]. Arthroplasty surgery has life-changing benefits, as these surgeries relieve pain and restore function [94]. However, a significant number of prosthetic joints fail due to biofilm-related infections that are difficult to treat. The incidence of prosthetic joint infection in the US was 2.18% of the total number of hip and knee arthroplasties performed in 2009 and has been estimated to increase over time [95]. While infections of orthopedic devices carry a low attributable mortality rate, the economic burden of treatment is substantial [96].

*S. aureus* is the most common cause of prosthetic joint infections, with studies reporting the involvement of this bacterium in up to 57% of total prosthetic joint infections [97]. The mode of infection may be direct inoculation during surgery or the hematogenous route [98]. If the bacterium is inoculated during surgery, it causes acute infection within 3 months; however, the infection may also occur at any time after surgery through the hematogenous route [99]. A low number of bacteria, such as <50 CFU of *S. aureus*, are enough to establish infection, in comparison to 10^4^ CFU in the absence of an implant [100].

Surgical interventions for prosthetic joint infections are debridement with polypropylene liner exchange and one- or two-stage re-implantation operations—all followed by prolonged antimicrobial therapy [98]. While the success rates are in the range of 85–90%, there are a proportion of patients who are either not suitable for surgery or in whom these costly procedures fail [98,101,102]. Additionally, all surgical procedures have considerable morbidity [98]. Novel adjuvant treatments that may be able to eradicate prosthetic joint infections would be highly beneficial.

## 7. Possible Adjuvant Treatments for Biofilm-Related *S. aureus* Prosthetic Joint Infections—The Search for a Novel Approach to an Intractable Problem

We focus our discussion on treatment strategies such as QS inhibitors that target biofilm regulators and have already been used successfully to treat biofilm-related infection in animal models.

### 7.1. Quorum Sensing and Quorum Sensing Inhibitors: Stopping the Bacterial Communication

Quorum sensing includes a series of events, such as signal production, signal detection, and gene activation/inactivation [103], and results in group behaviors such as expression of virulence factors including biofilm [104]. Interruption of any steps of QS leads to failure in quorum sensing and has a detrimental effect on bacterial pathogenicity [103]. QS inhibiters do not directly kill bacteria; rather, they repress signal generation, block signal receptors, and disrupt the QS signal [103]. Therefore, there is less selection pressure and a low rate of resistance development, although it should be remembered that also, in this case, the possibility of the emergence of resistant mutants still exists [103,105]. However, there are no data on the dosage, route of administration, bioavailability, pharmacodynamic/pharmacokinetic profile, and toxicity of quorum sensing inhibitors, particularly in relation to their use in the treatment of prosthetic joint infection. Thus, further studies are needed. Non-peptide small molecules, peptides, and proteins are three main classes of QS inhibitors [103].

#### 7.1.1. RNAIII-Inhibiting Peptide

RNAIII-inhibiting peptide (RIP), alone or in combination with antibiotics and antimicrobial peptides, inhibits *S. aureus* biofilm, including biofilm formation by MRSA and glycopeptide-intermediate strains, and also helps in the treatment of established biofilms [106,107,108,109]. This peptide was efficient in the treatment of central venous catheter-associated infection, polymethylmethacrylate subcutaneous implant infection, and graft infection in animal models [106,107,108,109]. Synthetic RIP analogues and RIP derivatives have also similar activity as RIP [110,111]. However, not all RIP derivatives that inhibit RNAIII in vitro show efficacy for the inhibition of in vivo infection but only that containing lysine at position 2 and isoleucine at 4 4 [110]. This indicates that the activity of RIP derivatives depends upon the positioning of the amino acids, which gives a special spatial structure and property to the derived molecules, making them active even inside living beings. Further, even for closely related molecules that show similar activity in vitro, it is not guaranteed that they will also show similar activity in vivo.

RIP disarms *S. aureus* of its virulence factors by inhibiting RNAII and RNAIII—the two *agr* transcripts [110]. Due to structural homology, RIP competes with RNA III activating protein (RAP), a protein responsible for RNAIII synthesis, and prevents the phosphorylation of its target (TRAP) [112]. Vaccination using RAP was effective in the prevention of *S. aureus* infection in a cutaneous infection mouse model, as antibodies to RAP block the activation of RNAIII [111]. However, RIP/RAP/TRAP system analysis showed no evidence for its involvement in virulence determinant regulation, challenging the related findings [113]. Since the efficacy of RIP for the treatment of *S. aureus* infections is already established in animal models, further studies are needed to confirm the mechanism of action [106,107,108,109]. The RIP concentrations used in the animal studies were extremely high in comparison with effective native inhibitory AIP concentrations [114]. Additionally, RIP has been shown to reduce *S. aureus* adherence through *agr*-independent gene regulation [115]. Given that the antibacterial effect of RIP has never been tested, it may be that RIP has a direct, non-specific, inhibitory effect on *S. aureus* [114]. Although RIP, RIP derivatives, and RAP have been effective in the treatment and prevention of biofilm-related *S. aureus* infection in some other animal models, they are yet to be tested in prosthetic joint infection animal studies.

#### 7.1.2. Hamamelitannin

Hamamelitannin, a non-peptide analogue of the quorum sensing inhibitor RNAIII-inhibiting peptide (RIP), prevents in vitro as well as in vivo biofilm formation in *S. aureus*, including MRSA, by inhibiting attachment [116]. Hamamelitannin prevented infection in a subcutaneous graft rat model [116]. This plant-derived compound, when used in combination with vancomycin or clindamycin, shows a synergistic effect to remove the biofilm and increase host survival [117]. Hamamelitannin inhibits the *traP* QS system by interfering with its receptor and makes *S. aureus* biofilm more susceptible to vancomycin treatment [118]. At the molecular level, hamamelitannin alters the expression of genes involved in cell wall synthesis and eDNA release such that the increase in cell wall thickness and eDNA release induced by vancomycin treatment is inhibited [118]. Hamamelitannin and its analogues are good antibiotic potentiators, having been used successfully in treating a mouse model of *S. aureus* mastitis [119]. However, hamamelitannin and its analogues are yet to be tested in *S. aureus* prosthetic joint infection animal models.

#### 7.1.3. Auto-Inducing Peptides

AIPs are able to inhibit *agr* in multiple strains, making these molecules good candidates for the development of an anti-quorum sensing strategy. The *agr* expression of group I *S. aureus* is inhibited by group IV *S. aureus* supernatant but not vice versa [120]. However, synthetic AIPs of *agr* group I and group IV inhibited the *agr* expression of one another [121]. This discrepancy in the results may be due to a difference in the purity of the AIPs used in the two studies. In addition, the inhibitory role of a synthetic AgrDII peptide on subcutaneous abscesses caused by group I *S. aureus* strains has already been reported [122]. The thiolactone moiety gives a cyclic structure to AIP and is required for both biological activities, self-activation and cross-group inhibition, of AIP [121,122]—synthesized linear peptides (*agr* group II and III peptides, and RIP) that lack the thiolactone moiety are inactive [121]. However, modification of the AIP tail inhibits *agr* activation but not cross-group inhibition, implying the existence of different mechanisms for activation and inhibition [122]. Therefore, AIP or AIP analogues modified by tail removal or switching the position of rings and tails can be used as *agr* inhibitors [121,123,124]. Group II and I thiolactone peptide without tail represses all four groups of *S. aureus agr* [121,123]. The alanine-modified AIP group I and II, AIP group II lactone, and lactone analogues are QS inhibitors that do not act as activators for any *agr* groups [121,122]. Immunogenic challenge with cyclic peptide or analogue carried on a macromolecule can activate the humoral response against the native AIPs [125]. Antibodies produced thus, such as antibody against AIP-IV, quench the QS system [126]. Antibodies against AIP-IV prevented *S. aureus* subcutaneous infection in a mouse model and protected mice from a lethal intraperitoneal *S. aureus* infection [126]. In conclusion, *agr* quorum sensing can be inhibited either by preventing the accumulation of AIP or using the cross-group inhibition property of AIP, however, by neutralizing its self-group activation activity. Synthetic AgrDII peptides and antibodies against AIP-IV are yet to be tested in prosthetic joint infection animal studies. Other AIP-related molecules described under this topic that have shown effectiveness in in vitro studies are yet to be tested in animal models.

#### 7.1.4. Savirin

Savirin is a small synthetic molecule that, when injected subcutaneously, can both inhibit and treat *S. aureus* skin and subcutaneous infections in mouse models [127]. This molecule inhibits the attachment of AgrA to promoter regions, subsequently inhibiting the *agr* quorum sensing system and key virulence factors [127]. Thus, savirin disarms *S. aureus*, making it less competent to survive inside the host, which is subsequently cleared by the immune system [127]. Due to savirin’s low molecular weight (368), lipophilicity, and lack of reported animal model toxicity, this molecule meets all the criteria for an ideal agent for drug development [127]. Additionally, as savirin is a synthetic molecule, it could be synthesized in large quantities in pure form. Study of its biological activity could include structural modifications to improve savirin’s efficacy. Since the mode of action of savirin involves disarming bacterial virulence factors rather than direct inhibition, *S. aureus* does not appear to develop resistance to savirin as readily as it does against antibiotics [127].

Savirin may also be effective in the treatment of prosthetic joint infection as similar immune defense mechanisms relying on macrophages that are present in the skin also exist in joints [128]. However, higher doses or different sites of injection that ensure higher bio-availability at the site of infection may be needed, as it is difficult for drugs to penetrate through bones or joints [129]. In addition, the pharmacokinetics and pharmacodynamics of this molecule are not known. Therefore, further study is required to optimize the route of administration and dose for the treatment of prosthetic joint infection.

### 7.2. Drug Repurposing: Can Old Become New Again?

Drug repurposing relates to the use of existing or abandoned drugs for the treatment of diseases for which they were not originally developed [130]. Cheaper and faster clinical translation along with known safety profiles and pharmacology of existing drugs are the main advantages of drug repurposing [131]. Here, we discuss the drugs that have already shown efficacy in the treatment of biofilm-related *S. aureus* infection in animal models. We also include drugs with significant in vivo anti-biofilm activities, whose modes of action are yet to be known.

#### 7.2.1. Auranofin

Auranofin, an anti-arthritis drug, and its derivative, MH05, showed a positive effect in the treatment of biofilm-related *S. aureus* infection in an intra-peritoneal polypropylene mesh implant infection mouse model [132]. Auranofin’s antibacterial effect is through the inhibition of multiple key pathways responsible for the synthesis of important cell components, such as cell wall, DNA, and proteins [133]. Auranofin and MH05 did not eradicate the infection and mono-therapy with them may not be sufficient to treat biofilm-related infections mainly in immune-compromised patients [132]. Auranofin has been reported to show a significant synergistic effect with antibiotics, linezolid and fosfomycin, for the treatment of MRSA and MSSA cutaneous abscesses in a mice model [134]. Thus, adjuvant therapies using auranofin in combination with antibiotics may be beneficial to eradicate biofilm-related infection. Additionally, the emergence of auranofin-resistant *S. aureus* mutants is uncommon [135]. This drug may also be effective in the treatment of prosthetic joint infection; however, dose optimization would be required. Additionally, the mode of action of these drugs for the treatment of biofilm-related infections is not known and requires further study.

#### 7.2.2. Aspirin

Aspirin is among the most widely used drugs for its preventive effect on cardiovascular disease. In a catheter-induced *S. aureus* endocarditis rabbit model, aspirin treatment reduced bacterial biofilm, bacteremia, and, consequently, embolism [136]. Similarly, hemodialysis patients with tunneled catheters, when treated with aspirin, are less likely to develop an *S. aureus* blood infection [137]. Aspirin activates *sigB*, a stress-induced operon, and inhibits α-hemolysin (*hla*) and fibronectin (*fnbA*) gene expression [138]. *sigB* activation represses *sarA* and *agr* [138]. However, salicylic acid, the active component of aspirin, has also been reported to induce PIA-dependent *S. aureus* biofilm formation in a nasal colonization murine model using the Newman strain [139]. Environmental stresses such as acidic pH and salt content of nasal secretion may also contribute to increased biofilm formation [140]. Salicylic acid stabilizes in vitro *S. aureus* biofilm through *agr* quorum sensing system inhibition [141]. These results indicate that the *S. aureus* biofilm-related activity of aspirin is either strain-dependent or biofilm-type-dependent. Due to differences in the composition of colonizing materials in internal nares and heart valves, the mechanisms of biofilm formation and the types of biofilm formed may be quite different at the two locations. Heart valves are coated with plasma and may favor fibrin biofilm formation [68].

#### 7.2.3. Ticagrelor

Ticagrelor is an anti-platelet drug that is used for the treatment of atherosclerosis. It is a platelet adenosine diphosphate P2Y12 receptor inhibitor [142]. Post hoc analysis of large cardiovascular disease prevention studies showed that in acute coronary syndrome and pneumonia patients treated with ticagrelor, lower risks of infection-related death and improved lung function were present [143,144,145]. Subsequent investigation of this unexpected effect showed that ticagrelor inhibited *S. aureus* biofilm growth and bacterial dissemination to surrounding tissue in a pre-contaminated subcutaneous foreign body *S. aureus* infection mouse model [146]. However, the mode of action for the inhibition of biofilm formation is yet to be studied, but it can be hypothesized that ticagrelor downregulates key biofilm-related genes. Additionally, its antimicrobial effect may have contributed to the inhibition of biofilm formation, as this requires bacterial concentrations to reach a threshold level [147]. In vitro experiments using ticagrelor showed a synergistic effect with rifampicin, ciprofloxacin, and vancomycin [146]. The anti-MRSA antimicrobial activity of ticagrelor alone was superior to vancomycin but similar to daptomycin [146]. However, anti-platelet drugs, such as ticagrelor and aspirin, reduce the effect of platelet antibacterial peptides against *S. aureus*, when used alone or in combination [148].

#### 7.2.4. Simvastatin

The lipid lowering statin class of drugs have been tested for their antimicrobial activities, with simvastatin shown to have activity against *S. aureus* [149]. Simvastatin also inhibits biofilm formation by *S. aureus* and is more potent than linezolid or vancomycin in the disruption of established in vitro *S. aureus* biofilms [149,150]. Simvastatin reduced the bacterial burden in a murine MRSA skin infection model significantly [150]. This drug is known to inhibit adhesion, reduce cell viability, and lower extra-polysaccharide production [150]. However, molecular mechanisms for the inhibition of *S. aureus* biofilms are yet to be studied [150]. Additionally, there are no data available on the activity of simvastatin for treatment of prosthetic joint infection animal models.

#### 7.2.5. Thioridazine

Thioridazine, an antipsychotic drug, inhibited the dissemination of epicutaneously inoculated MSSA to the spleen and kidney and reduced abscess size produced by intradermally injected MSSA and MRSA [151]. This drug at its sub-inhibitory concentration enhanced the in vitro bactericidal effect of β-lactam antibiotics to MRSA [152]. However, the enhancement was not seen in a cutaneous abscess mouse model [151]. The thioridazine concentration required to reverse the methicillin resistance of MRSA used in the later study might be too high to attain in an animal model [151]. Thioridazine downregulates biofilm pathway genes, such as genes related to cell membrane and cell wall component synthesis and their transport, that are induced by *saeRS* and disturbs peptidoglycan biosynthesis [152]. This drug is yet to be tested in an *S. aureus* prosthetic joint infection animal model. However, since thioridazine has significant toxicity, the development of less toxic derivatives or significantly lower doses using adjuvant therapies would be beneficial.

## 8. Conclusions and Future Perspectives

Biofilm-related *S. aureus* prosthetic joint infections cause significant morbidity and, as treatment relying on surgical debridement and antibiotics is not universally effective, there is growing interest in the development of novel therapies. In this review, we highlight both novel molecules and repurposed drugs that have shown efficacy in the treatment of biofilm-related *S. aureus* infections in pre-clinical studies. In most cases, biofilms occurring in animal models of prosthetic joint infection have not been studied.

While repurposed drugs have defined pharmacokinetics, pharmacodynamics, and toxicity profiles, these are not available for the novel molecules described here. Additionally, the modes of action of biofilm disruption of many of the described novel molecules and drugs are still unknown and require further investigation. As the world’s population ages, there is an increasing reliance on the use of prosthetic joints. Arthroplasty surgery is among the most life-enhancing of all modern medical treatments. Failure of prosthetic joints due to infection requires broad consideration of novel treatment strategies.

## Figures and Tables

**Figure 1 antibiotics-10-01060-f001:**
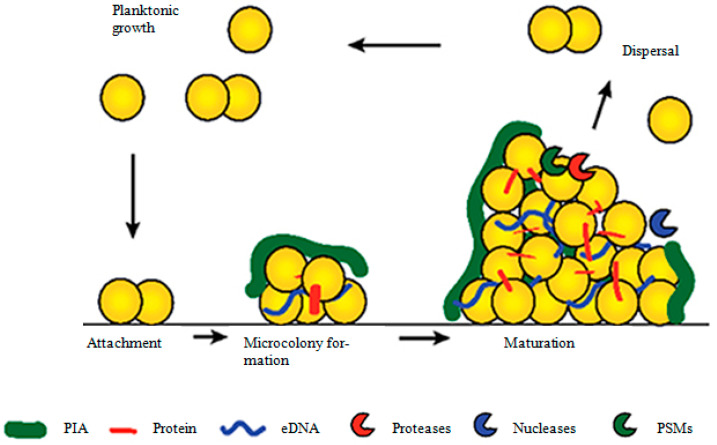
Biofilm growth cycle. Briefly, planktonic cells attach to surfaces and multiply to form micro-colonies. Micro-colonies then produce extracellular matrix and mature into a solid three-dimensional biofilm structure. After full maturation, extracellular matrix degrades and releases part of bacteria to establish a new biofilm at another location. (Adapted from https://www.frontiersin.org/files/Articles/123319/fcimb-04-00178-HTML/image_m/fcimb-04-00178-g001.jpg (accessed on 5 April 2021) by Lister J.L., Horswill A.R., *Staphylococcus aureus* biofilms: recent developments in biofilm dispersal, Frontiers in cellular and infection microbiology, 2014, by permission of copyright holders Lister and Horswill, 2014, under creative commons license http://creativecommons.org/licenses/by/4.0/ (accessed on 5 April 2021) [1]).

**Figure 4 antibiotics-10-01060-f004:**
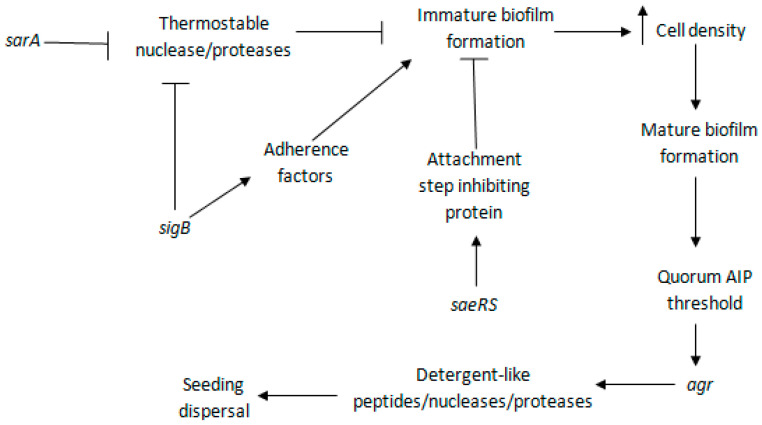
Regulation of *S. aureus* biofilm formation. Briefly, *sarA* and *sigB* inhibit expression of thermostable nuclease and proteases. *saeRS* upregulates expression of attachment step inhibiting protein. These enzymes and proteins inhibit immature biofilm formation. *sigB* upregulates expression of adherence factors that promote immature biofilm formation, leading to increased cell density and mature biofilm formation. When AIP accumulation reaches threshold level, *agr* is activated, which expresses detergent-like peptides, nucleases, and proteases, leading to biofilm dispersal. (Adapted from https://www.ncbi.nlm.nih.gov/pmc/articles/PMC3322633/figure/F5/?report=objectonly (accessed on 5 April 2021) by Archer N.K., Mazaitis M.J., Costerton J.W., et al., *Staphylococcus aureus* biofilms: properties, regulation, and roles in human disease, Virulence, 2011, by permission from copyright holders Taylor & Francis, 2011, under creative commons license https://creativecommons.org/licenses/by-nc/3.0/ (accessed on 5 April 2021) [71]).

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
