# Peer review of "Non-Antimicrobial Adjuvant Strategies to Tackle Biofilm-Related Staphylococcus aureus Prosthetic Joint Infections"

_antibiotics, 2021, doi:10.3390/antibiotics10091060_

Round 1
Reviewer 1 Report
In this review Authors performed comprehensive description of biofilm formation regulation and role of non-antimicrobial adjuvants in this process. Overall work is very interesting, and well written. I have some small comments included below:
- Line 10: Not only immune cells, but whole immune system response against bacteria presence.
- Lines 34-35: those stages should be indicated on the Figure 1.
- Figure 1: Please provide Figure with better quality. Further, most of the description of the Figure should be only included in Reference section. Please provide short description of biofilm formation instead.
- Line 60: MRSA please provide abbreviation expansion.
- Lines 60-61, 226-227, 486: S. aureus should be in one line (please change through the whole manuscript).
- Line 63: please indicate that C1qR is a receptor.
- Lines 99-101: Those are so called “moonlighting proteins”. Please expand this section a bit giving examples of those proteins. I don’t think putting information only about autolysis is a good idea, as there are a lot of hypothesis about that.
- Figure 2: the same comments as for Figure 1.
- Line 137: biofilm not Biofilm, do not end the title with “:”
- Line 171: Hlb abbreviation expansion is provided in line 168.
- General: Are there any other environmental conditions (despite iron depletion), that may impact on biofilm formation In S. aureus? That would be interesting to give an example of conditions typical for nasal ecological niches.
- Line 185: UAMS-1 please change the font size.
- Line 198: “…. and fibrin are other type of PIA..” The sentence becomes more clear.
- Line 200-201: at line 73 iron was nutrient, now its not, please standardize. Overall 200-202 lines are not clear.
- Line 210: Spa abbreviation was already extended in line 165. There are several mistakes like that through whole manuscript. Please standardize.
- Line 219-221: please rewrite this sentence, it is not clear.
- Line 255-258: please rewrite this sentence, it is not clear.
- Figure 3: same comments as for all Figures, further I think it should be divided into 2 Figures. E.g. Figure 3C should be placed close to the Section, and A and B closer to 3 and 4 section.
- Line 317: QS abbreviation should be placed earlier in the manuscript. Overall I think description of QS should appear earlier in the manuscript, as it was earlier described. As well as 7.1.3. AIP
- Lines 408-416, 465-475: please change size of the font.
- Please standardize the Reference section. Once its doi included once not etc.
Author Response
Dear sir/madam,
Thank you very much for reviewing my manuscript despite your busy schedule. Your valuable comments were really very helpful to improve my manuscript. The point to point responses to your comments are as follows:
1. Line 10: Not only immune cells, but whole immune system response against bacteria presence.
This sentence has been rewritten as you suggested.
2. Lines 34-35: those stages should be indicated on the Figure 1.
Done.
3. Figure 1: Please provide Figure with better quality. Further, most of the description of the Figure should be only included in Reference section. Please provide short description of biofilm formation instead.
Better quality figure has been provided. Short description of biofilm formation has been provided. The figures used in this paper were reproduced or adapted from already published papers, which I have indicated near the title of the picture as per standard rule.
4. Line 60: MRSA please provide abbreviation expansion.
Done.
5. Lines 60-61, 226-227, 486: S. aureus should be in one line (please change through the whole manuscript).
Done.
6. Line 63: please indicate that C1qR is a receptor.
Done.
7. Lines 99-101: Those are so called “moonlighting proteins”. Please expand this section a bit giving examples of those proteins. I don’t think putting information only about autolysis is a good idea, as there are a lot of hypothesis about that.
This section has been expanded and other hypotheses have been mentioned.
8. Figure 2: the same comments as for Figure 1.
Done.
9. Line 137: biofilm not Biofilm, do not end the title with “:”
Done.
10. Line 171: Hlb abbreviation expansion is provided in line 168.
Now, in line 171 only abbreviation has been used.
11. General: Are there any other environmental conditions (despite iron depletion), that may impact on biofilm formation In S. aureus? That would be interesting to give an example of conditions typical for nasal ecological niches.
Examples have been given about the low pH and salt content of nasal secretion which may enhance biofilm formation.
12. Line 185: UAMS-1 please change the font size.
Done.
13. Line 198: “…. and fibrin are other type of PIA..” The sentence becomes more clear.
Corrected as suggested.
14. Line 200-201: at line 73 iron was nutrient, now its not, please standardize. Overall 200-202 lines are not clear.
Done.
15. Line 210: Spa abbreviation was already extended in line 165. There are several mistakes like that through whole manuscript. Please standardize.
Done.
16. Line 219-221: please rewrite this sentence, it is not clear.
Done.
17. Line 255-258: please rewrite this sentence, it is not clear.
Done.
18. Figure 3: same comments as for all Figures, further I think it should be divided into 2 Figures. E.g. Figure 3C should be placed close to the Section, and A and B closer to 3 and 4 section.
Done.
19. Line 317: QS abbreviation should be placed earlier in the manuscript. Overall I think description of QS should appear earlier in the manuscript, as it was earlier described. As well as 7.1.3. AIP
Done.
20. Lines 408-416, 465-475: please change size of the font.
Done.
21. Please standardize the Reference section. Once its doi included once not etc.
Done.
Reviewer 2 Report
The aim of the review article entitled „ Non-Antimicrobial Adjuvant Strategies to Tackle Biofilm Related Staphylococcus aureus Prosthetic Joint Infections” was to present the current knowledge on the ability of S. aureus to form biofilms, with a special attention paid to prosthetic join infections, and briefly describe potential methods of combating them. I believe that the article is written carefully and in an understandable way. I noticed a few issues that still need to be corrected/ added.
Below I would like to present a list of amendments, the introduction of which, in my opinion, will improve the quality of the manuscript:
- „… are caused by the colonising nasal strain” -> … are caused by strains colonizing nasopharynx [line 24]
- “… while sessile forms cause chronic infections” -> … while sessile forms cause mainly chronic infections [line 25; as I believe they may sometimes produce acute infections]
- “… releases bacteria to establish a new biofilm at another location” -> … releases part of bacteria to establish a new biofilm at another location [line 38, some part of the biofilm for sure will stay in the area of the primary colonization]
- Figure 1: no legend of the structures present in the picture, please add this
- Section 2.1 AND Lines 206-207 AND Line 302: please change the citation placement so that it is consistent with the rest of the manuscript, i.e. at the end of a sentence or at the end of a given fragment of a sentence – not on the beginning
- Line 185 AND Lines 409-416 AND Lines 465-475: please reduce the font size according to the rest of the manuscript
- “… successfully to treat biofilm related infection animal models.” -> … successfully to treat biofilm related infections in animal models. [lines 315-316]
- “Therefore, there is less selection pressure and low rate of resistance development [102].” -> I cannot fully agree with the above sentence, so please modify it and quote the review presented below: Therefore, there is less selection pressure and low rate of resistance development, although it should be remembered that also in this case the possibility of the emergence of resistant mutants still exists […]
Krzyżek, P. Challenges and Limitations of Anti-Quorum Sensing Therapies. Front. Microbiol. 2019, 10, 2473.
- Section 7.2.1. Auranofin: this section should be expanded or deleted as it currently contains too little information
- I believe that it would be very beneficial for the readers and citation potential to add a table to the article, in which biofilm aureus components and their functions would be briefly discussed (proposal of columns: biofilm stage, biofilm/cells’ component, function(s), reference)
Author Response
Dear sir/madam,
Thank you very much for reviewing my manuscript despite your busy schedule. Your valuable comments were really very helpful to improve my manuscript. The point to point responses to your comments are as follows:
- „… are caused by the colonising nasal strain” -> … are caused by strains colonizing nasopharynx [line 24]
Done. In the study i have cited, they have mentioned anterior nares as the colonizing site for S. aureus. So, this sentence has been re-written accordingly.
- “… while sessile forms cause chronic infections” -> … while sessile forms cause mainly chronic infections [line 25; as I believe they may sometimes produce acute infections]
Done.
- “… releases bacteria to establish a new biofilm at another location” -> … releases part of bacteria to establish a new biofilm at another location [line 38, some part of the biofilm for sure will stay in the area of the primary colonization]
Done.
- Figure 1: no legend of the structures present in the picture, please add this
Done.
- Section 2.1 AND Lines 206-207 AND Line 302: please change the citation placement so that it is consistent with the rest of the manuscript, i.e. at the end of a sentence or at the end of a given fragment of a sentence – not on the beginning
Done.
- Line 185 AND Lines 409-416 AND Lines 465-475: please reduce the font size according to the rest of the manuscript
Done.
- “… successfully to treat biofilm related infection animal models.” -> … successfully to treat biofilm related infections in animal models. [lines 315-316]
Done.
- “Therefore, there is less selection pressure and low rate of resistance development [102].” -> I cannot fully agree with the above sentence, so please modify it and quote the review presented below: Therefore, there is less selection pressure and low rate of resistance development, although it should be remembered that also in this case the possibility of the emergence of resistant mutants still exists […]
Krzyżek, P. Challenges and Limitations of Anti-Quorum Sensing Therapies. Front. Microbiol. 2019, 10,
2473.
Done.
- Section 7.2.1. Auranofin: this section should be expanded or deleted as it currently contains too little information
This section has been expanded.
- I believe that it would be very beneficial for the readers and citation potential to add a table to the article, in which biofilm aureuscomponents and their functions would be briefly discussed (proposal of columns: biofilm stage, biofilm/cells’ component, function(s), reference)
Done. Table 1 has been added.
Reviewer 3 Report
The manuscript entitled “ Non-Antimicrobial Adjuvant Strategies to Tackle Biofilm Related Staphylococcus aureus Prosthetic Joint Infections” by Pant et al talks about S. aureus biofilms and describes various stages of their formation in great detail. It also gives a piece of very good information about the possible use for prevention and eradication of biofilm-related S. aureus prosthetic joint infection. Overall, the study is clear and concise. The introduction is relevant and theory-based. Sufficient information about the present study rationale and procedures are provided for the readers. The methods are generally appropriate, although clarification of a few details is required. Overall, the results are clear and compelling. The authors make a systematic contribution to the research literature considering the fact that antimicrobial resistance and biofilm development is a great cause of concern to all the clinical settings around the world. I find the manuscript suitable for publication in the journal after the minor comments below are addressed
Specific comments follow.
- Reference 1 in the introduction is a very old report from 1997. Can the authors find the latest report supporting their statement?
- Likewise, try to add more references from the years 2020 and 2021. I did not find any. This will be important to show that you have consulted literature deeply while writing your manuscript.
Please remake Figure 3. It seems as if the image was copied and the text “saeRS” and“Attachment step inhibiting protein” were pasted onto it.
Author Response
Dear sir/madam,
Thank you very much for reviewing my manuscript despite your busy schedule. Your valuable comments were really very helpful to improve my manuscript. The point to point responses to your comments are as follows:
- Reference 1 in the introduction is a very old report from 1997. Can the authors find the latest report supporting their statement?
Done. Now we have used a 2014 reference to support our statement.
- Likewise, try to add more references from the years 2020 and 2021. I did not find any. This will be important to show that you have consulted literature deeply while writing your manuscript.
Done. The references 5, 135 and 142 are from 2020 and 2021.
- Please remake Figure 3. It seems as if the image was copied and the text “saeRS” and“Attachment step inhibiting protein” were pasted onto it.
Done.